# Validity and Reliability of the Korean Version of the Self-Care of Diabetes Inventory (SCODI-K)

**DOI:** 10.3390/ijerph182212179

**Published:** 2021-11-19

**Authors:** Seung-Yeon Kong, Mi-Kyoung Cho

**Affiliations:** Department of Nursing Science, Chungbuk National University, Cheongju 28644, Korea; rrhd2@naver.com

**Keywords:** diabetes, self-care, self-efficacy, validity, instrument development

## Abstract

This was a methodological study to evaluate the validity and reliability of the Korean version of the Self-Care of Diabetes Inventory (SCODI-K). A total of 210 patients with type 2 diabetes from a university hospital were enrolled. Content validity, construct validity, and criterion-related validity were evaluated. Cronbach’s α was used to assess reliability. The SCODI-K consisted of 40 items in four dimensions (self-care maintenance, self-care monitoring, self-care management, and confidence). Four factors (activity-nutritional behavior, health-adherence behavior, health-promotion behavior, diet-restriction behavior) in the dimension of self-care maintenance, two factors (health status monitoring, symptom recognition) in the dimension of self-care monitoring, three factors (glucose self-control, problem-solving behavior, consultative self-care) in the dimension of self-care management, and one factor (self-care confidence) in the dimension of confidence were extracted. Confirmatory factor analysis supported a good fit with reliable scores for the SCODI-K model (normed chi-square(*χ²*/*df)* < 5, root mean square error of approximation (RMSEA) < 0.1, comparative fit index (CFI) ≥ 0.9, goodness-of-fit-index (GFI) ≥ 0.9). The SCODI-K showed a high positive correlation coefficient of 0.75 with the summary of diabetes self-care activities (SDSCA), confirming convergent validity. Cronbach’s α was 0.92 for the overall scale and 0.69 to 0.90 for the four dimensions. Therefore, the SCODI-K is a valid and reliable instrument for assessing self-care of patients with type 2 diabetes in Korea.

## 1. Introduction

Type 2 diabetes accounts for 90% of all diabetic patients. It is a representative chronic disease and a progressive disease, with decreasing insulin secretion. It is caused by aging, lack of exercise, obesity, being overweight, Westernized eating habits, and genetic factors [1]. The global diabetic population is estimated to be about 463 million and its prevalence is expected to increase rapidly [2]. In Korea, the prevalence of diabetes among adults aged over 30 years increased from 8.9% in 2001 to 14.4% in 2016. In adults aged over 65 years, its prevalence was 29.8%. It has been predicted that its prevalence will continue to increase because of aging and lifestyle changes [3].

Diabetes mellitus has a high disease burden. It is a chronic disease that is difficult to cure. Thus, it must be managed by controlling blood sugar throughout life. Neglect of management can lead to problems of large vessels and micro vessels, resulting in various and serious complications such as diabetic neuropathy, retinopathy, nephropathy, cardiovascular diseases, and cerebrovascular diseases [4]. It has been reported that 20–40% and 34.6% of diabetic patients have nephropathy and retinopathy, respectively. In addition, 40 to 60 million people have diabetic foot problems. Sarcopenia has also been reported in diabetic patients [2,5]. Since prognosis of diabetic patients greatly influences the occurrence of complications [2], patients need to perform regular self-care along with medical treatment for active blood-sugar control to prevent complications.

According to the Korean Diabetes Association [3], 24.1% of diabetic patients are smokers and 15.0% are high-risk drinkers. On the other hand, 36.0% of diabetic patients practice regular walking. The continuous treatment rate for diabetes is around 50%. Only about a quarter (25.1%) of diabetic patients have good blood-sugar control and only 10% of patients have blood-sugar levels, blood pressure, and cholesterol levels that are well regulated to target levels [3]. Thus, the importance of active and integrated management for diabetic patients has emerged. The importance of self-care to maintain a healthy life by means of steady management in daily life is being emphasized [2,3]. Self-care of diabetics requires patients to follow therapeutic instructions provided by medical staff, such as injection prescriptions, diet, medicine, exercise, self-performed blood-glucose tests, and foot care that can affect the regulation of blood glucose level in the patient’s own life to achieve the goal of treatment [6]. According to previous studies, self-care can significantly reduce HbA1c (glycosylated hemoglobin) and improve disease prognosis [7]. However, diabetic patients are stressed by the psychological burden of having to do self-care in their daily lives and the complex and difficult self-care guidelines. It has been reported that 70% of diabetic patients experience depressive symptoms. These negative emotions have been shown to worsen patients’ self-care in terms of proper eating habits and exercise, aggravate the disease, and increase complications [8,9,10,11] Therefore, the American Association of Diabetes Educators (AADE) has suggested that self-efficacy is a prerequisite for diabetic patients to properly perform self-care [12]. Self-efficacy means having confidence in being able to successfully carry out an action to achieve a certain outcome [13]. It has been found that increasing self-care adherence can have a positive effect on the change and maintenance of good health, decrease HbA1c, and ameliorate depression and diabetes distress [9,11,14,15]. Preventing complications and improving the quality of life are the most important goals of self-care for patients with diabetics to continuously monitor their own conditions with confidence to handle and solve problems such as hypoglycemia and hyperglycemia by themselves. Therefore, a tool that comprehensively includes the psychological area, problem-solving ability, and prevention of complications is needed. One can evaluate and understand self-care practically. Self-care results are related to physiological variables. Thus, it is necessary to actively evaluate the self-caring of diabetic patients using tools that are well documented for correlation.

Currently, most tools used for evaluating the self-care of diabetics are focused on self-care areas such as exercise, diet, blood-sugar monitoring, and medication. In addition to these existing self-care areas, American association of diabetes educators 7 systems (AADE7) has newly proposed problem-solving areas related to blood-sugar control (hypoglycemia, hyperglycemia, sick days), reducing risk factors for diabetes complications (by check-ups, smoking cessation, foot care, weight and blood-pressure control, vaccinations), and psychosocial adjustment (social support, stress management) necessary for continuous diabetes management [12]. It has been reported that the effect of self-care behavior is maximized when it is integrated rather than individual [16].

The Self-Care of Diabetes Inventory (SCODI) tool is based on the middle-range theory of chronic diseases. It can evaluate a patient’s self-caring with specific questions, including solving problems related to blood-sugar control in diabetic patients, reducing risk factors for complications, and self-care of patients. It was developed based on physiological variables [17]. Its validity and reliability have been verified.

The advantages of SCODI are described in detail as follows. First, SCODI is the first theory-based instrument for measuring the self-care behaviors of diabetic patients [18]. Second, SCODI was developed using the most recent clinical recommendations. It shows a high content validity index. There are at least 16 tools for assessing self-care of diabetes [18]. Of these, the most commonly used are the summary of diabetes self-care activities (SDSCA) and self-care inventory-revised version (SCI-R) [19,20]. They were developed before 2005. Third, psychological factors affecting the self-care of diabetic patients in previous studies are included in the self-confidence domain of SCODI [9,11,12,14,15,21]. Fourth, SCODI includes disease prevention behaviors (such as alcohol consumption, smoking habits, hand hygiene, and vaccination) and problem-solving ability to maintaining self-care. These factors are consistent with both the middle-range theory and standards of self-management education for diabetes. These are described as a basis for avoiding complications, disease exacerbations, and serious comorbidities [22]. Finally, although the original SCODI was developed in Italy, sociodemographic characteristics and clinical profiles of enrolled participants of the original study were international subjects [23], suggesting that the SCODI might be useful in other contexts. In addition, since the first use of SCODI in Italy in 2017, the feasibility of using this tool in various countries and regions such as Europe, China, North America, and South America has been actively validated.

Therefore, in this study, the Korean version of the Self-Care of Diabetes Inventory (SCODI-K) was developed. Its validity and reliability for diabetic patients in Korea were evaluated. The applicability of this tool was also explored.

## 2. Materials and Methods

### 2.1. Aims

To evaluate the validity and reliability of the Korean version of the Self-Care of Diabetes Inventory (SCODI-K).

### 2.2. Study Design

This was a methodological and cross-sectional study to evaluate the reliability and validity of SCODI (Self-Care of Diabetes Inventory, Monza, Italy), a Korean version of the self-care evaluation tool for diabetic patients originally developed by Ausili et al. [17].

### 2.3. Participants

Subjects of this study were type 2 diabetic patients who visited the outpatient endocrinology department of a university hospital located in Cheongju city, Korea. They were all 18 years of age or older who understood the purpose and method of this study. All subjects provided informed consent to participate in this study. They understood and filled out a questionnaire. The reliable and suitable sample size for factor analysis to confirm construct validity was 5 to 10 people per item [24]. We calculated the number of subjects based on the recommended sample size (200–400 people) [25] for confirmatory factor analysis using the AMOS program. Data were collected from 210 people and there were no dropouts.

### 2.4. Tools

#### 2.4.1. Characteristics of Participants

Demographic and sociological characteristics of subjects consisted of gender, age, marital status, education level, occupation, economic status, caregivers, alcohol, and smoking. Economic status was categorized into high, medium, and low levels. Characteristics related to the disease consisted of the duration of diabetes, complications of DM (diabetes mellitus), experience of hospitalization with DM, family history, experience of DM education, treatment modality, and comorbidity.

#### 2.4.2. Self-Care of Diabetes Inventory

We translated the SCODI (Self-Care of Diabetes Inventory), originally developed by Ausili et al. [17] for type 1 and type 2 diabetes patients based on the middle-range theory and clinical recommendations on self-care for chronic diseases. We verified the validity of experts and used it with the approval of the original author. This tool consisted of 40 questions in four dimensions: self-care maintenance (12 questions), self-care monitoring (8 questions), self-care management (9 questions), and self-care confidence (11 questions). On a 5-point Likert scale, self-care maintenance, self-care monitoring, and self-care management were rated from 1 (not at all) to 5 points (always do) and self-care confidence was rated from 1 (not confident about anything) to 5 points (can do everything well). The higher the score, the higher the degree of self-care. However, in the self-care maintenance area, item 29 was answered only by subjects who were taking insulin. Subjects who did not take insulin were excluded from the response. In addition, the score of the SCODI tool was evaluated as a score converted to 100 points. The score was converted based on 40 items for subjects taking insulin and on 39 items for subjects not taking insulin. At the time of development by Ausili et al. [17], the reliability of the tool (Cronbach’s α) was 0.81 for self-care maintenance, 0.84 for self-care monitoring, 0.86 for self-care management, and 0.89 for self-care confidence. In this study, Cronbach’s α was 0.77 for self-care maintenance, 0.68 for self-care monitoring, 0.74 for self-care management, and 0.90 for self-care confidence.

#### 2.4.3. The Summary of Diabetes Self-Care Activities (SDSCA) Questionnaire

We used the SDSCA developed by Toobert et al. [19] and translated into Korean by Chang and Song [26] to verify the criterion validity of the Korean version of SCODI. This tool used 17 questions to find out on how many days out of the last seven that five areas (diet, exercise, blood-glucose test, drugs, foot care) for diabetes management were carried out. Each question was answered on a scale from 0 point (not practiced on any day) to 7 points (doing it every day). The higher the score, the higher the degree of self-care for diabetes. At the time of development, the reliability of the tool (Cronbach’s α) was 0.68. The reliability of the Korean version of the tool was 0.77. In this study, it was 0.79.

#### 2.4.4. Physiological Indicators

In this study, body mass index (BMI), glycated hemoglobin (HbA1c), fasting blood glucose (FBG), and cholesterol were included as physiological indicators related to diabetes. The most recently measured results were collected from medical records. BMI was measured with a BSM330, InBody, Inc., Seoul, South Korea). HbA1c was measured with a glycosylated hemoglobin analyzer (HbA1c HA-8180, ARKRAY, Arkray, Inc., Kyoto, Japan). FBG and cholesterol were measured with an automatic chemical analyzer (TBA-FX8, Module 1, TOSHIBA, Toshiba, Inc., Tokyo, Japan).

### 2.5. Development of the Korean Version of the SCODI

The translation of the SCODI was completed by means of primary translation, expert review, reverse translation, and approval of the original author. First, before starting the primary translation, we obtained permission from the tool developer for translation and use of the tool in Korean. The primary translation of the SCODI tool was subcontracted to a specialized agency conducting several translations in the field. A preliminary translation was completed by a professor in the English Department fluent in English and Korean, who confirmed the consistency and accuracy of the original text. After that, the content validity of each item was verified twice by an expert group consisting of three professors of the Department of Endocrinology, one professor of the Department of Nursing, and one diabetes specialist. Whether each question was appropriate as a self-care question for diabetic patients and whether the expression and vocabulary were appropriate for translation into Korean were evaluated on a 4-point Likert scale (1 point, ‘very inappropriate’; 2 points, ‘inappropriate’; 3 points, ‘appropriate’; and 4 points, ‘very appropriate’). When marked as ‘very inappropriate’ or ‘inappropriate’, comments on correction and supplementation were written. After calculating the Item-Content Validity Index (I-CVI) of the five evaluators, we corrected the words and context of the sentence in the related question based on additional opinions for each question presented by the expert group for 13 items with CVI less than 0.80. The second content validity was confirmed by an expert group consisting of two professors in the Department of Endocrinology, one professor in the Department of Nursing, and one diabetes specialist. By calculating CVI, we found that all 40 items had CVI of 0.80 or more. After that, a reverse translation was requested from a specialized translation agency. The question was confirmed by validating the agreement between the reverse-translated question and the original tool. Finally, after we explained the process of translating the Korean version of the SCODI tool to the tool developer, the final translation was completed with confirmation that the tool was well translated into Korean by means of the translation and reverse-translation process of the developer (http://self-care-measures.com/available-self-care-measures/self-care-of-diabetes-inventory/, accessed on 12 May 2021) [27].

### 2.6. Data Collection

The data collection period for this study was from 1 November 2018 to 30 June 2019. Before collecting data, we explained the purpose and process of the study to nurses and professors of the Department of Endocrinology at the hospital and performed data collection after obtaining consent. Data collection was handled by a trained research assistant who was a fourth-year student in the Department of Nursing. This was conducted in an outpatient Department of Endocrinology. After explaining the purpose, the subjects, and contents of this study to an outpatient nurse of the Department of Endocrinology, we received a list of appointments and informed the research assistant after confirmation. Research assistants met with patients with type 2 diabetes both before and after their outpatient visits in order to collect data. The research assistant first revealed his/her identity, explained the purpose and procedure of the study to the patient, and obtained written consent. After explaining how to fill in the questionnaire, the assistant asked the subject to complete it. When the subject found it difficult to fill out the questionnaire because of blindness, the researcher or research assistant would read it and write down the answer after the subject responded. The average time required to complete the questionnaire was 15–20 min. A small gift was provided in return.

### 2.7. Ethical Consideration

This study was approved by the Institutional Review Board (IRB No. CBNUH 2018-10-020) of the hospital for the protection of subjects and ethical consideration of the study before beginning any research procedures. The study description stated the purpose of the study, the content of the study, the study procedure, guarantee of anonymity, and that the intention to participate in this study could be withdrawn and discontinued at any time even though the participant had consented to participate in this study. Contact information (telephone number, e-mail address) was provided so that a subject could contact us if there were any questions related to the study. Subjects who read the explanatory text and signed the consent to participate in the study wrote their answers down directly. All data were coded to maintain subject confidentiality.

### 2.8. Data Analysis

We analyzed the collected data using the IBM SPSS Amos and SPSS WIN 22.0 (IBM, Inc., Chicago, IL, USA) statistical programs. We obtained information about the demographic and sociological characteristics, health, and disease-related characteristics of subjects. Data are presented as percentage or mean and standard deviation. We verified the content validity using the Item-Content Validity Index (I-CVI). We verified construct validity by means of exploratory factor analysis and confirmatory factor analysis. For criterion validity, the correlation with the Summary of Diabetes Self-Care Activities (SDSCA) questionnaire was verified by Pearson’s correlation coefficients. The reliability of the Korean version of the SCODI was confirmed by its Cronbach’s α value.

## 3. Results

### 3.1. Characteristics of Participants and Their Difference in the SCODI-K

Altogether, 210 subjects participated in this study. Regarding demographic and sociological characteristics of subjects, 118 (56.2%) were males, 118 (56.2%) were over 60 years old, and 96 (45.7%) had a job. Regarding health and disease-related characteristics of subjects, 89 (42.4%) were alcohol drinkers, 32 (15.2%) were smokers, and 154 (73.3%) had a BMI of 23 kg/m^2^ or more. There were 102 (48.6%) who had a diagnosis of diabetes for more than 10 years. The most common diabetes treatment was taking oral hypoglycemic drugs by 162 (77.1%) patients. A total of 173 (82.4%) patients had comorbid diseases. HbA1c was 6.5% or higher in 164 (78.1%) patients. Fasting blood glucose was above 130 mg/dL in 112 (53.3%) patients and cholesterol was 200 mg/dL or higher in 24 (11.4%) patients (Table 1).

By analyzing the degree of self-care according to subject characteristics, we found that the self-care score was high when subjects were females (*t* = −3.75, *p* < 0.001), 60 years old or older (*t* = −4.93, *p* < 0.001), and unemployed (*t* = 3.66, *p* < 0.001). In addition, the self-care score was higher when subjects were non-drinkers (*t* = 2.70, *p* < 0.008) and non-smokers (*t* = 4.72, *p* < 0.001). The self-care score was higher when the diagnosis period was 10 years or more rather than less than 10 years (t = −4.27, *p* < 0.001). There was no difference in self-care scores according to other characteristics of subjects (Table 1).

### 3.2. Exploratory Factor Analysis of SCODI-K

#### 3.2.1. Self-Care Maintenance

The factor loading of 12 items in self-care maintenance ranged from 0.52 to 0.84. The following four factors explained 62.9% of the total variance: activity-nutritional behavior (Factor 1), health-adherence behavior (Factor 2), health-promotion behavior (Factor 3), and diet-restriction behavior (Factor 4). The explained variance by each factor was 10.4~19.7%. The total average score of self-care maintenance was 74.23 ± 16.47. Among items, ‘keep appointments with doctor’ had the highest score of 4.87 ± 0.55 while ‘exercise 2.5 h per week’ had the lowest score of 2.99 ± 1.60 (Table 2).

#### 3.2.2. Self-Care Monitoring

The factor loading of 8 items in self-care monitoring ranged from 0.47 to 0.90. Two factors explaining 56.8% of the total variance were extracted as follows: health status monitoring (Factor 1) and symptom recognition (Factor 2). The variance explained by each factor was 24.6~32.2%. Total average score of self-care monitoring was 66.01 ± 19.56. Among items, ‘pay attention to symptoms according to blood sugar’ had the highest score of 4.874.26 ± 1.15 and ‘recording blood sugar’ had the lowest score of 2.32 ± 1.62 (Table 2).

#### 3.2.3. Self-Care Management

Factor loading of 9 items in self-care management ranged from 0.51 to 0.84. Three factors explaining 66.6% of the total variance were extracted as follows: glucose self-control (Factor 1), problem-solving behavior (Factor 2), and consultative self-care (Factor 3). The variance explained by each factor was 24.6~32.2%. The total average score of self-care management was 56.56 ± 25.18. Among items, ‘adjust insulin dose per doctor’s suggestion’ had the highest score of 4.71 ± 0.65 while ‘recording cause of abnormal blood sugar’ had the lowest score of 2.54 ± 1.56 (Table 2).

#### 3.2.4. Self-Care Confidence

Self-care confidence was extracted with only one factor. The factor loading of its 11 items was 0.58~0.83. It explained 51.2% of the total variance. The total average score of self-care confidence was 79.15 ± 19.41. Among these items, ‘take medicine correctly’ had the highest score of 4.53 ± 0.87 while ‘prevent abnormal blood sugar’ had the lowest score of 3.82 ± 1.28 (Table 2).

### 3.3. Confirmatory Factor Analysis of SCODI-K

The CR value of four dimensions of *SCODI-K* was 0.9, which passed the standard of 0.5, indicating good reliability. The AVE of self-care maintenance was 0.4, which was slightly lower than the standard of 0.5. AVEs of the other three dimensions were 0.5, which passed the standard of 0.5, indicating good reliability (Table 2).

### 3.4. Construct Validity of SCODI-K

We found that the model fit of the four dimensions of SCODI did not meet the criteria for RMSEA, CFI, or GFI value overall. Thus, we constructed a modified model by checking the modification index in order to improve the fit of the model. The fit of the revised model satisfied standard values of χ^2^/df < 5 (range: 1.875~3.056), RMSEA < 0.1 (range: 0.065~.099), CFI ≥ 0.9 (range: 0.928~0.969), and GFI ≥ 0.9 (range: 0.912~0.963) in four dimensions (Table 3, Figure 1).

### 3.5. Criterion-Related and Concurrent Validity of SCODI-K

SCODI-K had a positive correlation with SDSCA (*r* = 0.75, *p* < 0.001). However, it showed no significant correlations with other variables. HbA1c had a positive correlation with fasting blood sugar (*r* = 0.56, *p* < 0.001) and cholesterol (*r* = 0.20, *p* < 0.004) (Table 4).

### 3.6. Reliability of SCODI-K

Reliability for all items of the SCODI-K in type 2 diabetes patients was 0.92. For the four dimensions of SCODI-K, reliability ranged from 0.69 to 0.90. The reliability of SDSCA was also 0.68 in the original SCODI. It was 0.79 in this study (Table 5).

## 4. Discussion

The purpose of this study was to develop a Korean version of the SCODI (Self-Care of Diabetes Inventory) recently developed in Italy to evaluate self-care behavior [17]. SCODI has an important effect on blood-sugar control and complication prevention in diabetic patients [28]. Thus, its validity and reliability were verified.

First, to verify the validity, we conducted exploratory factor analysis for the 40 items in the four dimensions of SCODI-K. Self-care maintenance, as in the original version and the Farsi version, consisted of four factors. However, there were differences in these items. The original version used the terms health-promoting exercise behaviors, disease prevention behaviors, health-promoting behaviors, and illness-related behaviors, whereas the Korean version used the terms activity-nutritional behavior, health-adherence behavior, health-promotion behavior, and diet-restriction behavior, which appeared as factors explaining the concept. These results show that concepts that can imply the item content of each factor are more specifically classified in the Korean version of SCODI. Self-care monitoring consisted of two factors, health status monitoring and symptom recognition. As in the original version, items were consistent except for No. 18. However, in the Farsi version [29], three factors were named, including symptom assessment in addition to symptom monitoring and symptom recognition, which showed differences. For self-care management, the original version named two factors. However, this study named three factors. These results show that subjects of this study considered autonomous taking care of oneself to include two categories: glucose self-control and problem solving. In particular, we found that ‘consultative’ was clearly classified, and that the result was similar to the Farsi version belonging to the same Asian region. For self-care confidence, the original version and the Farsi version showed two factors: specific self-care confidence and persistent self-care confidence. However, in this study, self-care confidence was named as one factor. Confidence is a perceived belief of an individual’s overall ability to perform successfully. It is believed that subjects of this study accepted the same meaning without distinguishing between specific and persistent.

Overall, the exploratory factor analysis (EFA) results showed differences from items included in each factor of the original version. These differences in results might be due to different cultural and medical environments of Italy and Korea, social and demographic characteristics, and differences between study subjects. However, even if the original version and factors were separated differently, factor loadings of all items were all over 0.40, indicating that the items of the tool explained the concept well with the same meaning as the original version. In this study, we performed CFA along with EFA to verify the validity of the tool, unlike with the original version or the Farsi version because the SCODI tool had already been validated. In order to apply it by translating it into Korean, we further secured the validity of the developed tool by conducting CFA, which confirmed the suitability of the item composition of extracted factors [30]. The four dimensions of 40 questions of the original version were applied as they were and implemented. The CFA showed that the CR values of all dimensions were above 0.90. AVE values were all above 0.50, except for self-care maintenance at 0.40. Thus, the validity was confirmed. It was difficult to compare with this study because CFA was not implemented at the time when the original version was developed. This result means that the content of each item reflects characteristics of subfactors well. In addition, the revised model’s fit and results were also found to satisfy all criteria. Therefore, it was clearly distinguished from items included in other factors as an appropriate item for measuring each corresponding factor.

To confirm the criterion validity of the Korean version of SCODI, we examined the correlation with the Summary of Diabetes Self-Care Activities (SDSCA) questionnaire tool developed by Toobert and Glasgow [19] with proven validity. SDSCA is a reliable tool that includes five categories of diabetes self-care: diet, exercise, drugs, foot care, and blood-sugar tests. The correlation coefficient between the two scales was 0.75, confirming that it would be an appropriate tool for measuring self-care in type 2 diabetes patients in Korea.

In this study, the total Cronbach’s α of the Korean version of the SCODI tool was 0.92. In the original version, direct comparison was difficult since only the reliability of the dimension was presented. However, it was higher than that of SDSC at 0.79. If the internal consistency was 0.70 or higher, it is an acceptable level. If it is 0.80 or higher, it means good reliability [31]. It has been proven to be a tool with high reliability for measuring the self-care of patients with type 2 diabetes in Korea.

The American Association of Diabetes Educators (AADE) [32] suggested seven areas of diabetes self-care activities (AADE7™ Self-Care Activities, the American Association of Diabetes Educators, Chicago, Illinois) as follows: healthy eating, being active, blood-glucose monitoring, taking medication, problem solving related to blood-sugar control, healthy coping, and reducing risks for diabetes complications. Self-care behavior has been shown to maximize the effect when it is conducted in an integrated rather than an individual way [16]. Self-care evaluation tools currently used by diabetic patients are focused on self-care areas that patients should carry out, such as exercise, diet, blood-sugar monitoring, and medication. Therefore, it is not possible to confirm the degree of empowering by which the patient can accept the disease and finally do self-care by solving problems that occur during self-care and nursing intervention. On the other hand, the SCODI tool is an evaluation tool consisting of 40 questions in four dimensions of self-care maintenance, monitoring, management, and confidence. It is composed of questions that can be specifically evaluated while including all seven areas suggested by AADE. Therefore, self-care performance can be predicted and evaluated more precisely. Based on this information, medical staff are expected to improve the effective self-care activity and problem-solving ability for each patient, thereby improving treatment outcomes, health conditions, and quality of life of patients.

When using a tool developed in a different language and culture, it must be used after thoroughly verifying its language validity, semantic equivalence, and conceptual equivalence through translation and reverse translation processes [33]. SCODI-K checked the content validity of the Korean situation through a diabetes expert group and confirmed whether patients understood the question well in a pilot test for diabetic patients [34]. Content validity was verified twice according to the Korean situation through a group of diabetes experts. The reliability and validity of the study results were verified and meaningful results were confirmed in a pilot study [34]. Therefore, the Korean version of the SCODI with validity and reliability verified in this study could be usefully employed as a tool to evaluate the self-care behavior of patients with type 2 diabetics in Korea beyond differences in cultural and medical environments from Italy, where the original tool was developed.

A limitation of this scale was the need for caution when calculating self-care scores. Because item 29 was the question, ‘If you find out that your blood sugar is too high or too low, do you adjust your insulin dosage in the way your health care provider suggested?’, only patients taking insulin should answer. To standardize the scale score, summing item responses, subtracting the number of items answered, and multiplying by the constant (http://self-care-measures.com/available-self-care-measures/self-care-of-diabetes-inventory/, accessed on 12 May 2021) [27] are needed.

## 5. Conclusions

Self-care by diabetic patients is very important to control blood sugar and prevent complications. Therefore, a tool to specifically evaluate self-care behavior is needed. While most tools currently used in Korea to assess self-care for diabetic patients focus on areas of self-care such as exercise, diet, blood-sugar monitoring, and medication, the Self-Care of Diabetes Inventory (SCODI) includes areas related to psychological aspects, problem-solving ability, and prevention of complications. Therefore, in this study, the Korean version of the SCODI was translated to verify its validity and reliability so that it could be applied to diabetic patients in Korea. In addition, the applicability of this tool was explored. Like the original version, the Korean version of the SCODI was composed of 40 questions in four dimensions of self-care maintenance, monitoring, management, and confidence. Suitability of the tool was verified through content validity, construct validity, criterion-related validity, and reliability. Although CFA was not implemented at the time the original version was developed, this study performed CFA with EFA to validate the SCODI. The results of this study show that the contents of each item adequately reflect the characteristics of subfactors.

## Figures and Tables

**Figure 1 ijerph-18-12179-f001:**
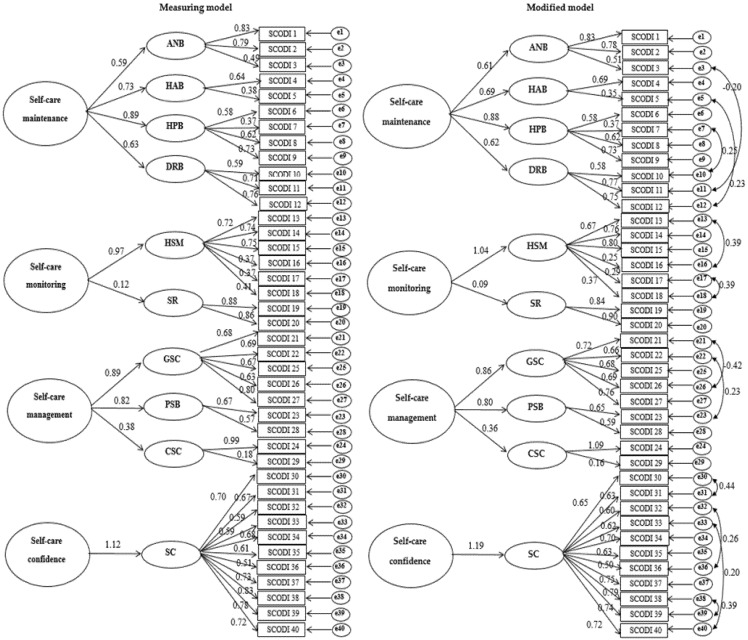
Measuring model and modified model of SCODI. Notes. SCODI: Self-Care of Diabetes Inventory; ANB: activity-nutritional behavior; HAB: health adherence behavior; HPB: health-promotion behavior; DRB: diet-restriction behavior; HSM: health status monitoring; SR: symptom recognition; GSC: glucose self-control; PSB: problem-solving behavior; CSC: consultative self-care; SC: self-care confidence.

**Table 1 ijerph-18-12179-t001:** Participants Characteristics and Difference in Self-Care According to Participants’ Characteristics (*N* = 210).

Characteristics	Categories	*N* (%)	Self-Care	
			M ± SD	t or F (*p*)
Gender	Male	118 (56.2)	66.27 ± 16.77	−3.75 (<0.001)
	Female	92 (43.8)	74.27 ± 14.13	
Age (year)	<60	92 (43.8)	63.88 ± 15.38	−4.93 (<0.001)
	≥60	118 (56.2)	74.37 ± 15.23	
Marital status	Married	180 (85.7)	70.04 ± 16.16	0.59 (0.556)
	The others	30 (14.3)	68.17 ± 16.15	
Education	≤Middle school	60 (28.6)	70.85 ± 17.52	0.61 (0.543)
	≥High school	150 (71.4)	69.35 ± 15.58	
Occupation	Unemployed	114 (54.3)	73.41 ± 14.50	3.66 (<0.001)
	Employed	96 (45.7)	65.46 ± 16.96	
Economic status	Low	33 (15.7)	65.30 ± 14.19	−1.74 (0.083)
	≥Middle	177 (84.3)	70.61 ± 16.37	
Care givers	No	73 (34.8)	68.27 ± 16.18	−0.99 (.326)
	Yes	137 (65.2)	70.58 ± 16.10	
Drinking	No	121 (57.6)	72.31 ± 14.94	2.70 (0.008)
	Yes	89 (42.4)	66.33 ± 17.10	
Smoking	No	178 (84.8)	71.90 ± 15.06	4.72 (<0.001)
	Yes	32 (15.2)	57.97 ± 17.00	
BMI (kg/m^2^)	<18.5	7 (3.3)	71.71 ± 16.17	0.90 (0.914)
	18.5~22.9	48 (22.9)	70.19 ± 19.03	
	≥23	154 (73.3)	69.48 ± 15.25	
Duration of disease (year)	<10	108 (51.4)	65.33 ± 16.91	−4.27 (<0.001)
	≥10	102 (48.6)	74.48 ± 13.85	
Complication of DM	No	101 (48.1)	69.19 ± 16.16	−0.51 (0.612)
	Yes	109 (51.9)	70.32 ± 16.16	
Experience of hospitalization with DM	No	158 (75.2)	69.69 ± 15.58	−0.14 (0.893)
	Yes	52 (24.8)	70.04 ± 17.86	
Family history	No	128 (61.0)	70.30 ± 16.05	0.58 (0.560)
	Yes	82 (39.0)	68.96 ± 16.31	
Experience of DM education	No	133 (63.3)	68.72 ± 16.22	−1.25 (0.214)
	Yes	77 (36.7)	71.60 ± 15.91	
Treatment modality	OHA	162 (77.1)	69.04 ± 16.42	1.51 (0.222)
	Insulin	36 (17.1)	70.61 ± 15.55	
	OHA and insulin	12 (5.7)	77.25 ± 12.59	
Comorbidity	No	37 (17.6)	70.51 ± 15.89	0.31 (0.760)
	Yes	173 (82.4)	69.62 ± 16.22	
HbA1c (%)	<6.5	46 (21.9)	68.63 ± 15.90	−0.54 (0.587)
	≥6.5	164 (78.1)	70.10 ± 16.23	
FBG (mg/dL)	<130	98 (46.7)	70.21 ± 15.81	0.37 (0.714)
	≥130	112 (53.3)	69.39 ± 16.46	
Cholesterol (mg/dL)	<200	186 (88.6)	69.53 ± 16.25	−0.62 (0.534)
	≥200	24 (11.4)	71.71 ± 15.36	

Notes. M: mean; SD: standard deviation; t: independent t-test; F: one-way analysis of variance; BMI: body mass index; DM: diabetes mellitus; OHA: oral hypoglycemic agents; HbA1c: glycosylated hemoglobin; FBG: fasting blood glucose.

**Table 2 ijerph-18-12179-t002:** Exploratory Factor Analysis and Confirmatory Factor Analysis of SCODI-K (*N* = 210).

Exploratory Factor Analysis						Confirmatory Factor Analysis	
Items	M ± SD	Factor Loadings					Factor Loadings
		1	2	3	4		
**Self-care**	68.03 ± 15.92						
**Self-care maintenance**	74.23 ± 16.47						
1. Do you maintain an active lifestyle? (Examples: walking, going out, doing outdoor activities.)	3.46 ± 1.40	0.83					0.83
2. Do you exercise 2.5 h per week? (Examples: swimming, going to the gym, riding a bicycle, walking.)	2.99 ± 1.60	0.80					0.79
3. Do you eat a balanced diet containing carbohydrates (pasta, rice, sugars, bread), protein (meats, fish, beans), fruits, and vegetables?	3.29 ± 1.32	0.64					0.49
12. Many people do not properly take their prescribed medicine. Do you take all medicines prescribed by your doctor?	4.74 ± 0.72		0.80				0.76
11. Do you get your health check-ups on time? (Examples: blood test, urine test, ultrasound, eye check-up.)	4.66 ± 0.87		0.79				0.71
10. Do you keep your appointments with your doctor?	4.87 ± 0.55		0.73				0.60
7. Are you refraining from smoking?	4.35 ± 1.44			0.74			0.37
9. Do you maintain good oral hygiene? (Brush teeth at least twice a day. Use mouthwash and dental floss.)	4.05 ± 1.21			0.72			0.73
6. Are you managing your health? (Examples: washing hands, receiving recommended vaccines.)	4.22 ± 1.12			0.60			0.58
8. Do you care for your feet? (Apply moisturizer after washing and drying feet. Wear proper socks.)	3.67 ± 1.45			0.58			0.62
5. Are you limiting alcohol intake? (Allowable alcohol consumption is two shots of soju a day for men and one shot of soju a day for women.)	4.12 ± 1.39				0.84		0.38
4. Do you avoid salt and fats? (Example: cheese, cured meats, sweets, red meats.)	3.18 ± 1.33				0.52		0.64
Eigenvalue		2.36	2.03	1.91	1.25		
Explained variance (%)		19.7	16.9	15.9	10.4	CR	0.9
Total explained variances (%)		19.7	36.6	52.5	62.9	AVE	0.4
**Self-care monitoring**	66.01 ± 19.56						
13. Do you regularly measure your blood sugar?	3.65 ± 1.44	0.80					0.72
15. Do you take your blood pressure?	3.31 ± 1.53	0.78					0.75
14. Do you check your weight?	3.91 ± 1.27	0.77					0.74
16. Do you keep a record of your blood sugar in a diary or notebook?	2.32 ± 1.62	0.52					0.37
17. Do you check your feet every day for wounds, redness, or blisters?	4.03 ± 1.37	0.48					0.37
18. Are you paying attention to high-blood-sugar symptoms (thirst, frequent urination) and low-blood-sugar symptoms (weakness, perspiration, anxiety)?	4.26 ± 1.15	0.47					0.41
19. Did you recognize quickly that you have the above symptoms?	3.52 ± 1.65		0.90				0.88
20. Did you recognize quickly that the above symptoms were caused by diabetes?	3.45 ± 1.71		0.90				0.86
Eigenvalue		2.58	1.97				
Explained variance (%)		32.2	24.6			CR	0.9
Total explained variances (%)		32.2	56.8			AVE	0.5
**Self-care management**	56.56 ± 25.18						
26. When you have high blood-sugar levels, do you lower your blood sugar by exercising?	3.40 ± 1.56	0.82					0.63
27. Do you check your blood sugar again to assess whether your actions to adjust the abnormal blood-sugar level were effective?	3.03 ± 1.66	0.80					0.80
21. Do you check your blood sugar when you feel you have symptoms (thirst, frequent urination, weakness, perspiration, anxiety)?	3.10 ± 1.60	0.75					0.68
25. When you have high blood-sugar levels, do you lower your blood sugar by adjusting your diet?	3.69 ± 1.42	0.66					0.67
22. When you have abnormal blood-sugar levels, do you keep a record of what caused it and what you did to address it?	2.54 ± 1.56	0.51					0.69
24. When you have low blood-sugar levels, do you solve the problem by eating sugary foods or drinks?	3.83 ± 1.44		0.76				0.99
29. When your blood-sugar levels are too high or low, do you adjust your insulin dose as per your doctor’s suggestion? *	4.71 ± 0.65		0.75				0.18
28. When your blood-sugar levels were too high or low, did you see your doctor?	3.28 ± 1.58			0.84			0.57
23. When you have abnormal blood-sugar levels, do you ask for advice from your family or friends?	2.92 ± 1.59			0.69			0.67
Eigenvalue		2.73	1.92	1.35			
Explained variance (%)		30.3	21.4	15.0		CR	0.9
Total explained variances (%)		30.3	51.6	66.6		AVE	0.5
**Self-care confidence**	79.15 ± 19.41						
38. I can take actions to control my blood-sugar level and relieve symptoms.	4.11 ± 1.15	0.83					0.83
39. I can evaluate whether the actions taken to control blood-sugar levels and relieve symptoms were effective.	3.90 ± 1.19	0.78					0.78
37. I can continue self-care for diabetes no matter what.	4.30 ± 1.01	0.76					0.73
40. I can continue carrying out those actions to improve my blood-sugar levels in any situation.	4.25 ± 0.94	0.75					0.72
34. I can check my blood-sugar level as many times as ordered by my doctor.	4.20 ± 1.12	0.73					0.68
30. I can prevent high blood sugar or low blood sugar and their symptoms.	3.82 ± 1.28	0.73					0.70
31. I can adhere to the nutrition and physical activity guidelines.	3.86 ± 1.18	0.71					0.67
33. I can persist to the treatment plan in any situation.	4.45 ± 0.89	0.67					0.61
32. I can take my medicine correctly. (Including insulin, as prescribed.)	4.53 ± 0.87	0.67					0.59
35. When I check my blood-sugar levels, I understand whether it is under control.	4.28 ± 1.07	0.66					0.61
36. I can recognize low-blood-sugar symptoms.	4.11 ± 1.28	0.58					0.51
Eigenvalue		5.64					
Explained variance (%)		51.2				CR	0.9
Total explained variances (%)		51.2				AVE	0.5

Notes. * Item 29, Only 48 patients receiving insulin responded to this question.; M: mean; SD: standard deviation; Evaluation criterion of standardized factor loading was 0.5; CR ratio was ≥0.7; AVE was ≥0.5; SCODI: Self-Care of Diabetes Inventory; AVE: average variance extracted; CR ratio: composite reliability; self-care maintenance factor 1: activity-nutritional behavior; self-care factor 2: health-adherence behavior; self-care factor 3: health-promotion behavior; self-care factor 4: diet-restriction behavior; self-care monitoring factor 1: health status monitoring; self-care monitoring factor 2: symptom recognition; self-care management factor 1: glucose self-control; self-care management factor 2: problem-solving behavior; self-care management factor 3: consultative self-care; self-care confidence factor 1: self-care confidence.

**Table 3 ijerph-18-12179-t003:** Model fit of SCODI-K (*N* = 210).

Variable	Measuring Model				Modified Model			
	χ²/df	RMSEA	CFI	GFI	χ²/*df*	RMSEA	CFI	GFI
SCODI								
Self-care maintenance	2.214	0.076	0.896	0.919	1.875	0.065	0.930	0.937
Self-care monitoring	4.771	0.134	0.855	0.900	1.915	0.066	0.969	0.963
Self-care management	3.447	0.108	0.874	0.912	2.491	0.084	0.933	0.945
Self-care confidence	4.753	0.134	0.855	0.846	3.056	0.099	0.928	0.912

Notes. Evaluation criteria for standardized χ²/*df* is <5, RMSEA is <0.1, CFI is ≥0.9, GFI is ≥0.9. SCODI: Self-Care of Diabetes Inventory; χ²/*df*: normed chi-square; RMSEA: root mean square error of approximation; CFI: comparative fit index; GFI: goodness of fit index.

**Table 4 ijerph-18-12179-t004:** Criterion-Related and Concurrent Validity of SCODI-K (*N* = 210).

Variables	SCODI	SDSCA	HbA1c	FBG
	r (*p*)			
SDSCA	0.75 (<0.001)			
HbA1c	0.04 (0.563)	0.11 (0.109)		
FBG	0.01 (0.993)	0.02 (0.757)	0.56 (<0.001)	
Cholesterol	0.02 (0.744)	−0.01 (0.921)	0.20 (0.004)	0.07 (0.293)

Notes. SCODI: Self-Care of Diabetes Inventory; SDSCA: Summary of Diabetes Self-Care Activities; HbA1c: glycosylated hemoglobin; FBG: fasting blood glucose.

**Table 5 ijerph-18-12179-t005:** Reliability of SCODI-K (*N* = 210).

Variables	Items	M ± SD	Cronbach’s α	
			Korean Version	Original
SCODI	40	69.78 ± 16.13	0.92	
Self-care maintenance	12	74.23 ± 16.47	0.77	0.81
Self-care monitoring	8	66.01 ± 19.56	0.69	0.84
Self-care management	9	56.56 ± 25.18	0.81	0.86
Self-care confidence	11	79.15 ± 19.41	0.90	0.89
SDSCA	17	37.08 ± 14.44	0.79	0.68

Notes. SCODI: Self-Care of Diabetes Inventory; SDSCA: Summary of Diabetes Self-Care Activities.

## Data Availability

The data are not publicly available because of privacy concerns.

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
