# Peer review of "Validity and Reliability of the Korean Version of the Self-Care of Diabetes Inventory (SCODI-K)"

_ijerph, 2021, doi:10.3390/ijerph182212179_

Round 1
Reviewer 1 Report
The narrative and process of the overall manuscript have value and significance, but the culture of South Korea is different from that of other countries such as Europe and the United States. The cultural applicability and limitations of this scale should be explained more clearly.
Author Response
"Please see the attachment."

Reviewer 2 Report
This article evaluated the validity and reliability of the Korean version of the Self-Care of Diabetes Inventory to provide a reliable tool for assessing self-care behaviors in Korean patients with diabetes. However, it seems that there were still some questions to be solved.
(1) The introduction described Type 2 diabetes messages in a large section, but as the object of the research, the information about the questionnaire was too brief and vague. What’s more, the comparison of this tool with other tools was only mentioned in the discussion part, which did not reflect the benefits of the tool and the need for the study. In short, this article is not structured well enough. Authors should focus on the questionnaire itself and present more details at the beginning, such as the source of the questionnaire as well as the necessity and advantages of choosing it. Also, the author should expand the discussion to cover these elements as appropriate.
(2) Line 95-96: Please provide references where their validity has been verified.
(3) Table 1: What’s the classification boundary for Economic status?
(4) Line 286: Numbering error, same below until line 310.
(5) At the end of the experiment, were there any improvements found?
(6) Duplicate use of references was found. Please reference more additional related and updated papers.
Author Response
"Please see the attachment."

Reviewer 3 Report
The authors have previously published an article titled "Factors Related to Self-care in Patients with Type 2 Diabetes" in 2020 and concluded that patients with self-efficacy had the best glycemic control. This paper seems to be a continuation of the previous study and shows the benefits of using the SCODI-K model for improving self-care in type 2 diabetic patients. The paper has been very nicely presented and the results very nicely explained. I congratulate the authors on completing such a nice study and I recommend that it moves forward for publication. I have only one minor comment regarding table 2 that the subheadings (Self-care maintenance, Self-care Monitoring, Self-care Management and Self-care confidence) would have bigger font as now they are only bold and with the same font size makes it difficult to read.
Author Response
"Please see the attachment."

Round 2
Reviewer 2 Report
Thanks for your response. This article is clear than before. But there was still a mistake in it.
Line 400: Misspelling. The word "Kore" should be corrected to "Korea".
Author Response
"Please see the attachment."
